# Intermittency of Arctic-midlatitude teleconnections: stratospheric pathway between autumn sea ice and the winter NAO

Peter Yu Feng Siew[1,2], Camille Li[1,2], Stefan Pieter Sobolowski[3,2], and Martin Peter King[3,2]

[1]Geophysical Institute, University of Bergen, Bergen, Norway
[2]Bjerknes Centre for Climate Research, Bergen, Norway
[3]NORCE, Bergen, Norway

**Correspondence:** Peter Yu Feng Siew (yu.siew@uib.no)

**Abstract.** There is an observed relationship linking Arctic sea ice conditions in autumn to midlatitude weather the following winter. Of interest in this study is a hypothesized stratospheric pathway whereby reduced sea ice in the Barents-Kara Seas enhances upward wave activity and wave-breaking in the stratosphere, leading to a weakening of the polar vortex and a transition of the North Atlantic Oscillation (NAO) to its negative phase. The Causal Effect Networks (CEN) framework is used to explore

the stratospheric pathway between late autumn Barents-Kara sea ice and the February NAO, focusing on its seasonal evolution, timescale-dependence, and robustness. Results indicate that the pathway is statistically detectable and has been relatively active over the 39-year observational period used here, explaining approximately 26% of the interannual variability in the February NAO. However, a bootstrap-based resampling test reveals that the pathway is highly intermittent: the full stratospheric pathway appears in only 16% of the sample populations derived from observations, with individual causal linkages ranging from

46 to 84% in occurrence rates. The pathway's intermittency is consistent with the weak signal-to-noise ratio of the atmospheric response to Arctic sea ice variability in modelling experiments, and suggests that Arctic-midlatitude teleconnections might be favoured in certain background states. On shorter time scales, the CEN detects two-way interactions between Barents-Kara sea ice and the midlatitude circulation that indicate a role for synoptic variability associated with blocking over the Urals region and moist air intrusions from the Euro-Atlantic sector. This synoptic variability has the potential to interfere with the strato-

spheric pathway, thereby contributing to its intermittency. This study helps quantify the robustness of causal linkages within the stratospheric pathway, and provides insight into which linkages are most subject to sampling issues within the relatively short observational record. Overall, the results should help guide the analysis and design of ensemble modelling experiments required to improve physical understanding of Arctic-midlatitude teleconnections.

## 1 Introduction

Autumn sea ice is a potential source of skill in predicting the winter North Atlantic Oscillation (NAO), and hence, European climate (Scaife et al., 2014; Wang et al., 2017; Hall et al., 2017). One proposed mechanism for the relationship focuses

on the Barents-Kara Seas, a region with seasonal ice cover that has exhibited strong negative trends during the cold season over the last decades (Cavalieri and Parkinson, 2012; Serreze and Stroeve, 2015; Onarheim and Årthun, 2017). According to this mechanism, reduced Barents-Kara sea ice triggers a wave response that constructively interferes with the climatological stationary wave pattern (Peings and Magnusdottir, 2014; Kim et al., 2014; Sun et al., 2015; Nakamura et al., 2016; Wu and Smith, 2016; Hoshi et al., 2017; Zhang et al., 2018a; De and Wu, 2018), enhancing upward propagation of planetary waves that weakens the stratospheric polar vortex (Nishii et al., 2009; Garfinkel et al., 2010; Smith et al., 2010). Downward coupling from the stratosphere to the troposphere subsequently produces circulation anomalies that resemble the negative phase of the NAO or Arctic Oscillation (AO) (Baldwin and Dunkerton, 1999; Polvani and Waugh, 2004), along with its attendant climate effects (Hurrell, 1995).

A delayed stratospheric pathway linking sea ice and the NAO is suggested by observations, but its exact nature is somewhat unclear. The observational evidence (e.g., García-Serrano et al., 2015; King et al., 2016; Koenigk et al., 2016) hinges on lagged correlations such as the one shown in Fig. 1a (similar to Fig. 10c in García-Serrano et al. (2015) and Fig. 6b in King et al. (2016)): less Barents-Kara sea ice in November is associated with higher polar cap heights in the stratosphere (i.e., polar vortex weakening), and a subsequent downward propagation of the height anomalies into the troposphere through the winter season, consistent with the appearance of negative NAO conditions several months later. However, the stationarity and statistical significance of this signal has been questioned when using longer records that extend back before the satellite era (Hopsch et al., 2012; Kolstad and Screen, 2019). In fact, the strength and timing of the signal can change when the observational period in Fig. 1a is extended by just several additional winters, showing a statistically insignificant autumn sea ice connection to the winter NAO via the stratosphere (Fig. 1b).

Evidence from modelling experiments is even more difficult to interpret because the relationship between Barents-Kara sea ice and the NAO is not robust in simulations. Some studies find a clear stratospheric signal after removing sea ice, leading to a weakening of the polar vortex and a negative NAO (Kim et al., 2014; Nakamura et al., 2015; Sun et al., 2015). A negative NAO response to sea ice loss is also possible, although much weaker, if the stratospheric pathway is not well represented or artificially suppressed (Liptak and Strong, 2014; Sun et al., 2015; Wu and Smith, 2016; Nakamura et al., 2016; Zhang et al., 2018a; De and Wu, 2018). However, other modelling studies show a weak or even positive NAO response when sea ice is reduced (Singarayer et al., 2006; Strey et al., 2010; Orsolini et al., 2012; Cassano et al., 2014; Screen et al., 2014), and we lack a comprehensive understanding of why model results are so different (Screen et al., 2018). One reason may be that the atmospheric response depends on where and when sea ice is removed; for example, some studies have shown that sea ice loss in the Pacific sector leads to a strengthening of the polar vortex (Sun et al., 2015; Screen, 2017; McKenna et al., 2018), and that winter ice loss may be more influential than autumn ice loss in weakening and shifting the jet stream (Blackport and Screen, 2019). Other possible reasons include nonlinearities with respect to the amplitude of sea ice loss (Petoukhov and Semenov, 2010; Semenov and Latif, 2015; Chen et al., 2016; Overland et al., 2016), and dependence of the atmospheric response on the background state (Smith et al., 2017, 2019; Labe et al., 2019).

Overall, isolating the sea ice influence on the midlatitudes remains a challenge in part because it is a search for causal drivers in a tightly coupled system with large internal variability (Shepherd, 2016). This internal atmospheric variability itself has

well-known effects on Arctic climate over a range of time scales. Synoptic weather systems carry heat and moisture poleward from the North Atlantic, and are associated with moist intrusions that have been shown to warm the Arctic and melt sea ice (Woods et al., 2013; Park et al., 2015a, b; Gong and Luo, 2017; Kim et al., 2017; Lee et al., 2017). Feedbacks between sea ice and the NAO acting on intraseasonal time scales can yield opposite-signed relationships depending on the time lag considered: anomalously low Barents-Kara sea ice concentrations are favoured by positive NAO conditions (Fang and Wallace, 1994; Deser et al., 2000), but are also part of an ice perturbation pattern that has been found to produce negative NAO conditions (Magnusdottir et al., 2004; Deser et al., 2004; Kvamstø et al., 2004; Strong et al., 2009; Deser et al., 2010; Wu and Zhang, 2010). The causality problem with respect to sea ice extends beyond the NAO to other midlatitude phenomena such as Eurasian cooling, for which one finds numerous studies arguing both for (Outten and Esau, 2012; Mori et al., 2014, 2019) and against (McCusker et al., 2016; Sorokina et al., 2016; Ogawa et al., 2018; Blackport et al., 2019) sea ice loss being responsible for the recent spate of extreme winters.

In the present study, we revisit the observed relationship between autumn Barents-Kara sea ice and the winter NAO with the goal of quantifying the robustness of the stratospheric linkage. In other words, we ask how systematically the stratospheric linkage has appeared during the satellite period. While sampling issues are unavoidable when using a short observational record with large internal variability, our analysis attempts to account for this by exploring the idea that weak but statistically significant signals may arise from a teleconnection pathway that is only intermittently active.

We begin with a description of data and methods (section 2), including a Causal Effect Networks (CEN) approach that provides a statistical framework for assessing causality (applied to climate problems by studies such as Ebert-Uphoff and Deng, 2012; Runge et al., 2014; Kretschmer et al., 2016, 2018). Results showing that the pathway is indeed detectable but exhibits a high level of intermittency are presented in section 3, and the implications for understanding present day Arctic-midlatitude teleconnections are discussed in section 4. We end with some concluding remarks in section 5.

## 2 Data and Methods

### 2.1 Reanalysis data

The Causal Effect Networks (CEN) approach requires indices (time series) of variables representing key processes in the dynamical mechanism being studied. In our study, we use sea ice area fraction, surface sensible heat flux, surface latent heat flux, sea level pressure, meridional wind, temperature, geopotential height, and downward thermal radiation at the surface. Raw daily data for the period 1979 to 2018 are from the European Center for Medium-Range Weather Forecasts (ECMWF) ERA-Interim reanalysis (Dee et al., 2011). The seasonal cycle is removed at each grid point by subtracting the climatological daily mean to obtain anomalies of each variable, and the data are detrended. The trend is removed through all the days of the year (1 January, 2 January, etc.). The following indices are then calculated from the reanalysis data from September to March:

- Barents-Kara sea ice (ICE): sea ice area fraction averaged over 70°-80° N, 30°-105° E (Fig. 2a)

- Barents-Kara turbulent heat flux (THF): sum of surface sensible and latent heat flux averaged over 70°-80° N, 30°-105° E (Fig. 2b), with positive defined as heat flux from the ocean to the atmosphere

- stratospheric polar vortex strength (SPV): negative of geopotential height poleward of 60° N (Fig. 2c) averaged between 10-100 hPa, as defined by Kretschmer et al. (2016), such that positive values of the index indicate a stronger polar vortex

- Urals sea level pressure (URALS): sea level pressure averaged over 45°-70° N, 40°-85° E (Fig. 2d)

- downward longwave radiation (IR): downward thermal radiation at the surface averaged over 70°-90° N (Fig. 2e)

- poleward eddy heat flux (V*T*): product of V* and T* at 100 hPa averaged over 45°-75° N (Fig. 2f), where V and T denote the meridional wind velocity and air temperature respectively, and the superscript * indicates deviations from the zonal mean

- North Atlantic Oscillation index (NAO): from the Climate Prediction Center, based on Rotated Principal Component Analysis of 500 hPa geopotential height, see details at
https://www.cpc.ncep.noaa.gov/products/precip/CWlink/pna/nao.shtml

Finally, the daily indices are averaged up to monthly, half-monthly and pentad means for the different analyses carried out in this study.

## 2.2 Causal Effect Networks (CEN)

The CEN algorithm is a causal inference framework (Runge et al., 2014, 2019) aimed at identifying causal relationships between variables of interest. It was previously used to study Arctic-midlatitude teleconnections by Kretschmer et al. (2016, 2018). Essentially, given a set of indices such as the ones described above, a CEN is constructed following three steps: 1) identify potential causal drivers of each index (condition selection), 2) identify the causal drivers using these potential causal drivers as a "conditioning set", and 3) quantify the strength of the causal relationship. We will illustrate the algorithm using January stratospheric polar vortex strength ($SPV_{Jan}$) as an example. Readers are referred to Kretschmer et al. (2018) and Runge et al. (2019) for a full description of the CEN algorithm, also known as PCMCI.

In the first step, we find all possible drivers for $SPV_{Jan}$. A preliminary list of drivers is generated by calculating the Pearson correlation $r$ between $SPV_{Jan}$ and all other indices (including SPV itself) in the preceding months, up to a maximum lag of 2 months (i.e., November and December for this example). Indices with significant correlations are retained, where an optimal significance level is determined using the Akaike information criterion (AIC). The AIC results in the selection of a 20% significance level for the case of $SPV_{Jan}$ (note that the AIC allows for these rather liberal significance levels in the first step, but more stringent levels are used later in the second step). This leaves us with the following possible drivers: $V*T*_{Dec}$, $URALS_{Dec}$, $SPV_{Dec}$ and $URALS_{Nov}$. This list is sorted in descending order according to the absolute value of the Pearson correlation coefficient. Next, we test for the conditional independence of all four possible drivers with $SPV_{Jan}$ by calculating partial correlations, controlling for the effect of each driver one at a time starting from the top of the sorted list. If a driver

passes the partial correlation test, it is retained in the list of possible drivers; if it does not pass, it is removed from the list, meaning it is no longer in the conditioning set. For example, the partial correlation between $\text{URALS}_{\text{Nov}}$ and $\text{SPV}_{\text{Jan}}$ controlling for $\text{V*T*}_{\text{Dec}}$ is, following the notation of Kretschmer et al. (2016):

$$\rho(\text{URALS}_{\text{Nov}}, \text{ SPV}_{\text{Jan}} \mid \text{V*T*}_{\text{Dec}}) = -0.274, \tag{1}$$

where

$$\rho(x, y \mid z) = \frac{r_{xy} - r_{xz} r_{yz}}{\sqrt{1 - r_{xz}^2} \sqrt{1 - r_{yz}^2}} \tag{2}$$

The partial correlation is significant at the 20% level (p-value = 0.105), therefore, $\text{URALS}_{\text{Nov}}$ is retained as a possible driver of $\text{SPV}_{\text{Jan}}$. After going through the entire list, $\text{SPV}_{\text{Dec}}$ is eliminated, leaving us with three possible drivers of $\text{SPV}_{\text{Jan}}$: $\text{URALS}_{\text{Nov}}$, $\text{URALS}_{\text{Dec}}$ and $\text{V*T*}_{\text{Dec}}$.

In the second step, we retest all possible links (for all indices in the preceding two months, including those rejected in the
130 first step) with $\text{SPV}_{\text{Jan}}$, controlling for the *combined effect* of the possible drivers (conditioning set) identified in the first step. This step helps account for false positives when working with highly interdependent time series (as is often the case with climate indices), and enhances detection power (Runge et al., 2019). Specifically, the test for $\text{SPV}_{\text{Jan}}$ is:

$$\rho(\text{X}, \text{ SPV}_{\text{Jan}} \mid \text{URALS}_{\text{Nov}}, \text{ URALS}_{\text{Dec}}, \text{ V*T*}_{\text{Dec}}), \tag{3}$$

where X represents all indices of ICE, THF, URALS, V*T*, SPV and NAO in both November and December. Any X producing
a significant partial correlation in Eq. 3 is regarded as a causal driver of $\text{SPV}_{\text{Jan}}$. The conditioning set excludes X when X is being tested, for example:

$$\rho(\text{V*T*}_{\text{Dec}}, \text{ SPV}_{\text{Jan}} \mid \text{URALS}_{\text{Nov}}, \text{ URALS}_{\text{Dec}}) = -0.453 \tag{4}$$

which is significant at the 5% level (p-value=0.00629). Testing all X leaves us with three causal drivers of $\text{SPV}_{\text{Jan}}$ : $\text{URALS}_{\text{Nov}}$, $\text{URALS}_{\text{Dec}}$ and $\text{V*T*}_{\text{Dec}}$. Note that these are the same causal drivers identified in the first step, meaning that no new drivers are
140 reintroduced in the second step in this case. As an additional refinement, the Hochberg-Benjamini false discovery rate (FDR) control may be used to account for the multiple testing problem (Kretschmer et al., 2018; Runge et al., 2019).

In the third step, we use a multiple regression equation to quantify the influence of causal drivers and simultaneous influences on $\text{SPV}_{\text{Jan}}$:

$$\text{SPV}_{\text{Jan}}^s = \beta_0 + \beta_1 * \text{URALS}_{\text{Nov}}^s + \beta_2 * \text{URALS}_{\text{Dec}}^s + \beta_3 * \text{V*T*}_{\text{Dec}}^s + \beta_4 * \text{Y}_{\text{Jan}}^s \tag{5}$$

where the $\beta$ values are regression coefficients for the standardized regressors $\text{URALS}_{\text{Nov}}^s$, $\text{URALS}_{\text{Dec}}^s$, $\text{V*T*}_{\text{Dec}}^s$, $\text{Y}_{\text{Jan}}^s$, and the superscript $s$ indicates a standardized index. The inclusion of Y allows us to check for significant simultaneous relationships between all indices. By standardizing, the interpretation is that changing a certain regressor by one standard deviation changes $\text{SPV}_{\text{Jan}}$ by $\beta$ standard deviations, provided that all other variables are held fixed.

A two-tailed t-test is used for significance testing. For the AIC in step one, a significance set of (5%, 10%, 20%) is used. There are no substantial changes to the main messages when using other significance sets (Fig. S4). A significance level of 5% is used in the second and third steps.

The above example illustrates how the CEN algorithm identifies and evaluates causal drivers of $SPV_{Jan}$. In order to construct the complete monthly and half-monthly CENs, we identify causal drivers for all our chosen indices (ICE, THF, URALS, V*T*, SPV and NAO) during the extended winter season (NDJFM). September to December (January to March) indices are taken over the period of 1979 to 2017 (1980 to 2018). All Pearson correlations and partial correlations (first and second steps) and the multiple regressions (third step) are thus based on indices with a sample size of 39 winter seasons. A similar procedure is used for the pentad CEN, but with a maximum lag of two pentads to capture processes occurring on synoptic time scales.

## 3   Results

This section describes results from our exploration of the ICE-NAO stratospheric pathway using the CEN framework (section 3.1), including an assessment of its strength (section 3.2) and intermittency (section 3.3) in the observational record. We also explore processes occurring on shorter timescales, and discuss how these effects may reinforce or interrupt the stratospheric pathway (section 3.4).

### 3.1   Seasonally evolving ICE-NAO pathway

We begin by examining pathways from Barents-Kara sea ice to the NAO proposed by previous studies. The CEN analysis follows the approach of Kretschmer et al. (2016), but keeps individual months separate rather than considering the DJF period as a whole. This allows us to capture the seasonal evolution of pathways through the cold season.

The CEN (Fig. 3) shows evidence for a stratospheric pathway leading from autumn sea ice perturbations in the Barents-Kara Seas to a late winter NAO response. This pathway appears using both monthly (Fig. 3a) and half-monthly (Fig. 3b) averages as input to the CEN, albeit with slight differences in timing. The half-monthly CEN in Fig. 3b is displayed such that individual half-monthly linkages (shown in Fig. S2) are aggregated into full months to allow for direct comparison to Fig. 3a.

Coloured arrows in the network diagrams highlight the ICE-NAO stratospheric pathway, where red indicates positive relationships and blue indicates negative relationships (the exact values correspond to the beta coefficients in the multiple regression equation, e.g., Eq. 5). Grey arrows show other linkages that are statistically significant, including some tropospheric pathways that also contribute to the ICE-NAO relationship. A figure including all the identified causal linkages and autocorrelations appears in the supplementary material (Fig. S1). For the monthly CEN, the stratospheric pathway is

$$\downarrow ICE_{Oct} \Rightarrow \uparrow URALS_{Dec} \Rightarrow \uparrow V^*T^*_{Dec/Jan} \Rightarrow \downarrow SPV_{Jan/Feb} \Rightarrow \downarrow NAO_{Feb/Mar}$$

where we use the notation A $\Rightarrow$ B to indicate index A as a "driver" of index B, and $\downarrow$ and $\uparrow$ to represent a decrease or increase, respectively, of the indices. The pathway is described for the case of a negative sea ice perturbation leading to a negative NAO.

For the half-monthly CEN, the pathway may be summarized as:

$\downarrow \text{ICE}_{\text{Oct/Nov}} \Rightarrow \uparrow \text{THF}_{\text{Nov}} \Rightarrow \uparrow \text{URALS}_{\text{Dec}} \Rightarrow \uparrow \text{V*T*}_{\text{Dec/Jan}} \Rightarrow \downarrow \text{SPV}_{\text{Dec/Jan/Feb}} \Rightarrow \downarrow \text{NAO}_{\text{Feb/Mar}}$

Using the finer half-monthly resolution in the CEN prevents shorter timescale processes (such as linkages through THF) from being averaged out.

The CEN results illustrate how the stratospheric pathway unfolds through the winter season. The timing is in general agreement with previous observational studies, suggesting that the involvement of the stratosphere introduces a few months' delay
in the NAO response to Barents-Kara sea ice variability (Kim et al., 2014; García-Serrano et al., 2015; Jaiser et al., 2016; King et al., 2016; Kretschmer et al., 2016; Yang et al., 2016). The causal linkages are consistent with the idea that Arctic sea ice reduction enhances upward wave activity through constructive interference between forced Rossby waves and the climatological stationary waves (Garfinkel et al., 2010; Smith et al., 2010). The resulting increase in wave-breaking in the stratosphere decelerates the polar vortex (Charney and Drazin, 1961), which in turn leads to tropospheric circulation anomalies and surface
impacts via downward coupling (Baldwin and Dunkerton, 1999). Some features of the pathway, such as the relatively long lagged relationship of autumn sea ice to December Urals sea level pressure, are not well understood, an issue that will be further discussed in section 4.

We will focus on the stratospheric pathway from $\text{ICE}_{\text{Oct}}$ to $\text{NAO}_{\text{Feb}}$ in the monthly CEN, as this timing yields the strongest negative ICE-NAO correlation (Fig. S5). The correlation between $\text{ICE}_{\text{Nov}}$ and $\text{NAO}_{\text{Jan}}$ is equally strong, but the causal pathway
goes through the troposphere only (Fig. S1b) and is not a focus of this study. Results from the half-monthly CEN yield consistent messages, and will be brought into the discussion where relevant.

## 3.2 Strength of the pathway

An interesting question is how to assess the strength of the ICE-NAO stratospheric pathway as a whole, and what insights may be gained by such an assessment.
The CEN analysis yields a set of beta coefficients (colours of the arrows in Fig. 3) that describe the strength of individual causal linkages in our network. Following Runge et al. (2015), the *total causal effect* of the stratospheric pathway from $\text{ICE}_{\text{Oct}}$ to $\text{NAO}_{\text{Feb}}$ may be calculated by summing over the product of beta coefficients along the two relevant chains of linkages from Fig. 3a:

$\downarrow \text{ICE}_{\text{Oct}} \xRightarrow{-0.326} \uparrow \text{URALS}_{\text{Dec}} \xRightarrow{0.390} \uparrow \text{V*T*}_{\text{Dec}} \xRightarrow{-0.368} \downarrow \text{SPV}_{\text{Jan}} \xRightarrow{0.426} \downarrow \text{NAO}_{\text{Feb}} \ (0.0199)$
$\downarrow \text{ICE}_{\text{Oct}} \xRightarrow{-0.326} \uparrow \text{URALS}_{\text{Dec}} \xRightarrow{-0.449} \downarrow \text{SPV}_{\text{Jan}} \xRightarrow{0.426} \downarrow \text{NAO}_{\text{Feb}} \ (0.0624)$

The total causal effect (0.0199 + 0.0624 = 0.0823) tells us that a one-standard deviation perturbation in $\text{ICE}_{\text{Oct}}$ yields a likesigned response of 8% of one-standard deviation in February NAO (Runge et al., 2015). One might question the interpretation of the contemporaneous $\uparrow \text{URALS}_{\text{Dec}} \Rightarrow \uparrow \text{V*T*}_{\text{Dec}}$ linkage in the first chain as a causal effect, but the fact that it also shows
up in the half-monthly CEN as a linkage from the first half of the December to the second half (Fig. S2) supports the point.

A comparison between the stratospheric and tropospheric ICE-NAO pathways shows that the latter are generally stronger in the CEN framework. Table 1 summarizes the causal effect of the three full pathways (Fig. S6). Our main stratospheric pathway of interest from $ICE_{Oct}$ to $NAO_{Feb}$ is comparable in strength to the pathway from $ICE_{Oct}$ to $NAO_{Mar}$ (0.0823 and 0.0872). The latter has both stratospheric and tropospheric chains, accounting for 30% (0.0258/(0.0614+0.0258)) and 70% (0.0614/(0.0614+0.0258)) of the total causal effect, respectively. The $\downarrow ICE_{Jan} \Rightarrow \downarrow NAO_{Mar}$ tropospheric pathway is the strongest in terms of the total causal effect (0.137), primarily because it involves fewer linkages. Overall, the larger causal effect of the tropospheric pathways is perhaps unsurprising, given that the stratospheric pathway may be disrupted by internal variability (noise) from both the troposphere and the stratosphere.

An alternative view of the pathway strength comes from considering the amount of February NAO variance explained by the various linkages along the pathway using a multiple linear regression framework. This gives a sense of the relative importance of each linkage, and how information passes through the pathway. The full pathway can be represented by the following regression equation:

$$NAO_{Feb} = \kappa_0 + \kappa_1 \cdot ICE_{Oct} + \kappa_2 \cdot URALS_{Dec} + \kappa_3 \cdot V^*T^*_{Dec} + \kappa_4 \cdot SPV_{Jan} \tag{6}$$

where $\kappa_0$ is a constant and $\kappa_1$, $\kappa_2$, $\kappa_3$, $\kappa_4$ are the regression coefficients for the standardized regressors $ICE_{Oct}$, $URALS_{Dec}$, $V^*T^*_{Dec}$ and $SPV_{Jan}$, respectively. The importance of the regressors may be quantified in different ways, for example:

a) cumulative $NAO_{Feb}$ variance explained as regressors are included, calculated by successively adding terms in Eq. 6 from left to right (orange bars in Fig. 4), e.g. for $V^*T^*_{Dec}$:

$$NAO_{Feb} = \kappa_0^a + \kappa_1^a \cdot ICE_{Oct} + \kappa_2^a \cdot URALS_{Dec} + \kappa_3^a \cdot V^*T^*_{Dec} \tag{7}$$

b) $NAO_{Feb}$ variance explained by individual regressors, calculated via a simple bivariate regression between each regressor and $NAO_{Feb}$ (blue bars), e.g., for $V^*T^*_{Dec}$:

$$NAO_{Feb} = \kappa_0^b + \kappa_3^b \cdot V^*T^*_{Dec} \tag{8}$$

c) reduction in $NAO_{Feb}$ variance explained when individual regressors are removed, calculated by removing the term from the regression equation (green bars), e.g., for $V^*T^*_{Dec}$:

$$NAO_{Feb} = \kappa_0^c + \kappa_1^c \cdot ICE_{Oct} + \kappa_2^c \cdot URALS_{Dec} + \kappa_4^c \cdot SPV_{Jan} \tag{9}$$

Both the blue and green bars in Fig. 4 provide a measure of the contribution of individual regressors, while comparison of these with the orange bars gives some indication of whether information from a given regressor is redundant.

The stratospheric pathway explains 26% of the variance in the February NAO (Fig. 4). The cumulative variance explained (orange bars) increases from 11% to 26% as regressors are added (moving from left to right), indicating that each linkage in the pathway adds some useful information. This result is consistent with other estimates from observations, but likely represents

an upper limit as the Barents-Kara sea ice and NAO relationship is shown to be particularly strong during the current reanalysis period compared to the rest of the twentieth century (Kolstad and Screen, 2019).

    While successive linkages in the pathway add explanatory power, they are not independent. Comparing the orange and blue bars, we see that the increase of cumulative explained variance moving from left to right is much less than the explained variance from each individual regressor. For example, while $SPV_{Jan}$ explains the most NAO variance of any individual regressor

(18%), its removal from the full regression does not have much effect (3% reduction in explained variance), However, we know that variability in upward wave activity and variability in the polar vortex are closely related, so in a sense, it is not physically meaningful to consider one in isolation of the other. Removing both $V*T*_{Dec}$ and $SPV_{Jan}$ from the regression equation results in a 8% reduction (not shown) in explained variance, which is perhaps a more representative estimate of the stratosphere's contribution. Sea ice appears to impart information that cannot be explained by the other three regressors (6% reduction in

explained NAO variance when removed), but this may also be a result of atmosphere-ice feedbacks explored in section 3.4.

    Overall, these analyses show a role for the stratosphere in connecting autumn ICE to late winter NAO, but one that accounts for a modest fraction of the total NAO variance. However, the pathway strength reported here should be considered as an estimate, as there remain uncertainties associated with analysis choices such as the time resolution of the input data and the relevant lags to include. In the next section, we will further explore reasons for this relatively weak ICE-NAO covariability.

### 3.3    Intermittency of the pathway

    The ICE-NAO stratospheric pathway identified by the CEN comprises statistical relationships inferred from a relatively short observational record of only 39 winters. It is meaningful to ask how robust the pathway is, that is, how systematically the relevant statistical relationships occur in the record. To assess the robustness, we perform a bootstrapping test, where bootstrap samples are created by randomly selecting 39 winters with replacement from the entire reanalysis period. The CEN of each

sample is then constructed. This procedure is repeated 10,000 times.

    The bootstrapping results (Fig. 5) indicate that the stratospheric pathway is intermittent. Percentages show the occurrence rate of individual segments in the pathway within the bootstrap sample population (see Fig. S7 for occurrence rates of other statistically significant linkages). By this measure, it is clear that individual segments have varying levels of intermittency, ranging from 46% for the segment $\downarrow SPV_{Jan} \Rightarrow \downarrow NAO_{Feb}$ to 84% for the segment $\uparrow V*T*_{Dec} \Rightarrow \downarrow SPV_{Jan}$. The full stratospheric

pathway (the sequence of all four segments) is detected in only 16% of the samples, suggesting that it does not occur systematically during every winter season. An alternative three-segment pathway $\downarrow ICE_{Oct} \Rightarrow \uparrow URALS_{Dec} \Rightarrow \downarrow SPV_{Jan} \Rightarrow \downarrow NAO_{Feb}$ is slightly less intermittent (22% occurrence rate), but its physical interpretation is unclear given that there is no linkage through $V*T*$ to the polar vortex, as expected from theory. These intermittency results are a likely reason why detection of the pathway is sensitive to the choice of observational period (Fig. 1), and suggests that it may be favoured in certain background states

(Overland et al., 2016; Smith et al., 2017).

    The existence of intermittency in the stratospheric pathway is consistent with previous suggestions that internal variability modulates the influence of Arctic sea ice on the midlatitude circulation (Screen et al., 2014; Overland et al., 2016; Shepherd, 2016). An examination of where in the pathway the intermittency is strongest provides clues to its origins. For example,

the upward coupling from sea ice to the stratosphere includes the segments $\downarrow \text{ICE}_{Oct} \Rightarrow \uparrow \text{URALS}_{Dec}$ and $\uparrow \text{URALS}_{Dec} \Rightarrow \uparrow$ $\text{V*T*}_{Dec}$, whose occurrence rates are 50% and 74%, respectively. The occurrence of these two linkages together is seen in about 41% out of 10,000 bootstrap samples, meaning that most of the time when the $\downarrow \text{ICE}_{Oct} \Rightarrow \uparrow \text{URALS}_{Dec}$ linkage is detected, the subsequent linkage to $\text{V*T*}_{Dec}$ follows. Conversely, when the $\uparrow \text{URALS}_{Dec} \Rightarrow \uparrow \text{V*T*}_{Dec}$ linkage is detected, it is preceded by the $\downarrow \text{ICE}_{Oct} \Rightarrow \uparrow \text{URALS}_{Dec}$ linkage in only about half the cases. An obvious source of the intermittency in both segments (individually and in terms of their "combined" occurrence rate) is regional SLP variability over the Urals related to atmospheric internal variability. Similarly, the downward coupling from SPV to NAO is vulnerable to both stratospheric and tropospheric internal variability, leading to a relatively low occurrence rate of 46%. This is consistent with the idea that not all polar vortex strengthening and weakening events affect the tropospheric circulation (Karpechko et al., 2017). Most robust is the $\uparrow \text{V*T*}_{Dec} \Rightarrow \downarrow \text{SPV}_{Jan}$ linkage (84%), which arises from well-known physical processes related to upward planetary wave flux and polar vortex weakening. Sea ice variability can also contribute to intermittency in the pathway through higher frequency synoptic processes, a topic we will explore in section 3.4.

The strength of the segments in the pathway also exhibits large variability among the bootstrap samples. This can be seen in histograms of the beta coefficients for all segments in the pathway (Fig. 5). While the beta coefficients exhibit ranges of up to 0.5 for any given segment, the sign is always the same, indicating that the sign of the relationship between variables is robust. The observed beta coefficients (black lines) for the reanalysis period itself fall within the spread of the distributions. Note that the distributions are composed only of samples in which the linkage of interest is detected by the CEN algorithm (i.e., a beta coefficient can be calculated from Eq. 5), which is why some of the distributions appear skewed. This is particularly true for the linkages that are least robust (the first and last segments, for which the observed beta coefficients are towards the weaker end of the distributions). Overall, these results indicate that even when the stratospheric pathway is active, there is substantial interannual variability in how it manifests.

### 3.4 Synoptic linkages and interactions across times scales

In the monthly CEN analysis, there are simultaneous relationships between Barents-Kara sea ice, Urals sea level pressure and the NAO (Fig. 6) that point to linkages through shorter timescale synoptic processes. For example, the NAO shows significant negative simultaneous relationships with Barents-Kara sea ice (positive NAO with reduced ice) in December and March, reflecting a well-known pattern of atmospheric forcing on sea ice via anomalies in surface heat fluxes driven by wind and temperature variability (Fang and Wallace, 1994; Deser et al., 2000). Additional simultaneous relationships between sea ice, turbulent heat flux, and Urals sea level pressure are consistent with synoptic features related to cyclones (Boisvert et al., 2016; Wickström et al., 2019) and moist intrusions (Woods et al., 2013; Park et al., 2015b) entering the Arctic. Moist intrusions in particular appear to occur preferentially during the positive phase of the NAO (Luo et al., 2017) and have been shown to lead to enhanced downward longwave radiation, surface warming, and sea ice reductions (Gong and Luo, 2017; Chen et al., 2018). We explore the possible influences of such events within the CEN framework by using higher frequency data to capture the relevant synoptic processes. The input data are pentad (5-day) means of Barents-Kara sea ice (ICE), Urals sea level pressure (URALS) and downward longwave radiation (IR). The maximum lag is set to two pentads (10 days) to isolate the synoptic timescale. The

results are summarized in Fig. 7 by summing the number of times a linkage appears in each month from Fig. S8. The maximum count for a given linkage in a month is 12 (six pentads in a month and up to 2-pentad lag considered). Autocorrelation is strong on these short time scales, and thus is not used to reject causal linkages in the partial correlation tests.

The CEN detects synoptic-scale influences from the Arctic to the midlatitudes that reinforce linkages found in the monthly analysis. A linkage from ICE to URALS appears regularly throughout the winter season (Fig. 7a), both indirectly through IR and as a direct connection, and in the correct sense to contribute to the $\downarrow \text{ICE}_{\text{Oct/Nov}} \Rightarrow \uparrow \text{URALS}_{\text{Dec}}$ linkage shown in the monthly and half-monthly CENs (Fig. 3). The $\downarrow \text{ICE} \Rightarrow \uparrow \text{IR}$ linkage (blue bars, first histogram in Fig. 7a) follows from the idea that sea ice retreat exposes open ocean, which is a local evaporative source for water vapour, leading to a moister, optically thicker atmosphere (Kim and Kim, 2017; Zhong et al., 2018). The linkage $\uparrow \text{IR} \Rightarrow \uparrow \text{URALS}$ (red bars, second histogram in Fig. 7a) is consistent with a suggested mechanism whereby the resulting surface warming weakens zonal wind locally and promotes blocking over the Urals (Luo et al., 2016). These synoptic processes, if habitually occurring, can imprint onto longer timescales, but may also produce interference effects, as seen by the appearance of opposite-signed causal relationships from those described above from time to time through the winter season.

At the same time, causal effects from the midlatitudes to the Arctic are also detected, consistent with an influence from moisture transport by cyclones or synoptic moist intrusions (Fig. 7b). This is represented by the $\uparrow \text{URALS} \Rightarrow \uparrow \text{IR}$ linkage (most frequently observed in October, January and February) and the $\uparrow \text{IR} \Rightarrow \downarrow \text{ICE}$ linkage (most frequently observed in November, January and February), which reflect the transport of moist air into the dry Arctic atmosphere by the large-scale flow or by cyclones tracking into the Barents. These midlatitude-to-Arctic linkages have a uniform sign (all red bars in first histogram, all blue bars in second histogram), suggesting that the effect of the relevant processes is rather systematic despite exhibiting month-to-month variability. We also detect a direct linkage from the Urals to Barents-Kara sea ice that can be of either sign. In the slightly more frequent negative sense ($\uparrow \text{URALS} \Rightarrow \downarrow \text{ICE}$), it can be interpreted as a direct effect of warm air advection and mechanical forcing of the ice cover from enhanced southerlies over the Barents-Kara region (Sorokina et al., 2016; McCusker et al., 2016). Together, these synoptic linkages show how Urals SLP variability, which has a large internally generated component, can reinforce or interrupt the ICE-NAO stratospheric pathway.

Given that our understanding of Arctic-midlatitude teleconnections must account for the combined influences of such linkages across regions and time scales, it is no surprise that we have yet to identify a definitive set of mechanisms. Implications of such scale interactions and how they relate to viewpoints presented in previous studies are further discussed in section 4.

## 4 Discussion

This study quantifies the robustness of atmospheric teleconnections between the Arctic and midlatitudes, documenting their high level of intermittency in the observational record. In a bootstrapping test, the full stratospheric pathway emerges in only 16% of the sample populations derived from the observations (Fig. 5). The existence of intermittency is likely why studies using various analytical approaches and time periods find teleconnections that differ in pattern, timing, robustness and apparent mechanisms (Overland et al., 2016; Francis, 2017; Cohen et al., 2018; Overland and Wang, 2018; Cohen et al., 2019). In this

section we discuss some of the factors that may contribute to the intermittency. Of course, anything that influences polar vortex strength is a potential source of intermittency (including internal variability, anthropogenic forcing, tropical variability, etc.), but we focus the discussion on factors that are most directly related to our CEN results.

To be more concrete, the intermittency of the stratospheric pathway stems from the fact that it can be reinforced or interrupted by other processes. For example, reinforcement can come from tropospheric pathways also detected by the CEN algorithm (see Fig. S1a and S1b):

   1)  $\downarrow ICE_{Oct} \Rightarrow \uparrow THF_{Nov} \Rightarrow \uparrow URALS_{Jan} \Rightarrow \uparrow URALS_{Feb} \Rightarrow \downarrow NAO_{Mar}$

   2)  $\downarrow ICE_{Jan} \Rightarrow \uparrow URALS_{Feb} \Rightarrow \downarrow NAO_{Mar}$

   3)  $\downarrow ICE_{Nov} \Rightarrow \downarrow NAO_{Jan}$

   4)  $\downarrow ICE_{Jan} \Rightarrow \downarrow NAO_{Mar}$

All these tropospheric and stratospheric pathways lead from the reduction of sea ice to a negative NAO, although they differ slightly in timing. The existence of the tropospheric pathway is supported by sea ice and surface heating perturbation experiments, where negative NAO/AO responses are simulated even when the stratospheric pathway is suppressed (Wu and Smith, 2016) or not well represented (Sun et al., 2015). However, the NAO/AO response is stronger when the stratospheric pathway is active than when it emerges through the tropospheric pathway alone (Nakamura et al., 2016; Zhang et al., 2018a, b).

Another example of a factor that may contribute to intermittency is the El Niño Southern Oscillation (ENSO). El Niño winters are associated with a deepened Aleutian low, which enhances upward propagating waves, weakens the polar vortex, and favours negative NAO conditions (Domeisen et al., 2019). As such, the stratospheric pathway may be reinforced if an El Niño develops following a low autumn ice season (both are associated with a weakened polar vortex, e.g., winter 1986/87 or 2009/10, Fig. 8); if a La Niña develops instead, the stratospheric pathway may be weakened (e.g., winter 2007/08 or 2010/11, Fig. 8). The relationship between wintertime ENSO and the NAO is rather weak (Brönnimann, 2007; Domeisen et al., 2019), consistent with Fig. 8, which shows high and low Nino3.4 values in both the lower (negative NAO) and upper (positive NAO) quadrants of the scatter plot. Given that we find no systematic phasing of ENSO with Barents-Kara sea ice variability during the reanalysis period, it is likely that ENSO contributes to intermittency in the ICE-NAO pathway.

In terms of reinforcing the stratospheric pathway, blocking over the Urals region seems to play a particularly important, but not fully understood, role. Enhanced Urals sea level pressure is closely linked to the Scandinavian pattern in Euro-Atlantic climate variability and is related, but not directly equivalent, to the occurrence of atmospheric blocking. The Urals linkage appears in the monthly CEN ($\downarrow ICE_{Oct} \Rightarrow \uparrow URALS_{Dec} \Rightarrow \uparrow V^*T^*_{Dec/Jan}$ in Fig. 3a). The latter segment from Urals sea level pressure to poleward eddy heat flux is fairly systematic (appears in 74% of the bootstrap samples in Fig. 5) and is grounded in the idea that tropospheric precursors over the Urals lead polar vortex weakening (Cohen and Jones, 2011; Cohen et al., 2014a). However, the first segment from Barents-Kara sea ice to Urals blocking is more intermittent (appears in 50% of the bootstrap samples), and whether it is in fact a causal linkage has been questioned by a recent modelling study using ensemble nudging experiments (Peings, 2019). Interestingly, not only Barents-Kara sea ice (Fig. 5e in King et al. (2016)) but also ENSO

(Figs 5e and 5f in King et al. (2018)) has been linked to the Scandinavian pattern, which suggests another avenue for ENSO to contribute to intermittency.

The ICE-URALS relationship highlights the complexity of interactions between atmospheric internal variability and Barents-Kara sea ice over a range of time scales. On synoptic scales, the pentad CEN (Fig. 7) shows linkages from reduced sea ice to enhanced Urals sea level pressure, but also linkages in the opposite direction ($\uparrow$ URALS $\Rightarrow \uparrow$ IR $\Rightarrow \downarrow$ ICE), with Urals sea level pressure altering ice cover via changes in poleward moisture transport that have been tied to synoptic moist intrusions (Woods et al., 2013; Luo et al., 2016; Gong and Luo, 2017; Lee et al., 2017). This chain of linkages can act as a positive feedback on sea ice perturbations, but also provides a pathway by which blocking variability (internal to the atmosphere) may interrupt the expected troposphere-stratosphere coupling in response to autumn sea ice (for example, imagine a case where atmospheric conditions inhibit Urals blocking after a low-ice autumn). Furthermore, enhanced Urals blocking and moist intrusions can lead to highly transient perturbations in turbulent heat flux over the Barents-Kara Seas. Initially, turbulent heat loss from the ocean is suppressed near the sea ice edge where moist intrusions act to weaken temperature and moisture contrasts between the atmosphere and ocean (Woods et al., 2013; Gong and Luo, 2017). But the heat flux anomaly can become positive (enhanced heat loss from the ocean) after the sea ice melts back in response to the moist intrusion, one to two weeks later (Woods and Caballero, 2016; Lee et al., 2017). On longer (monthly to seasonal) time scales, there is evidence that atmospheric variability is the main driver of heat flux variability over the Barents-Kara Seas both in observations and models (Sorokina et al., 2016; Blackport et al., 2019). This perhaps explains why turbulent heat flux does not show up in the monthly CEN (Fig. 3a), but does in the half-monthly CEN (Fig. 3b). Across synoptic to seasonal timescales, it appears that sea ice is best thought of as an intermediary rather than a true boundary forcing, as is implied by prescribed sea ice (e.g., AGCM) experiments.

One outstanding issue involves the mechanisms that have been proposed to explain the $\downarrow$ ICE$_{Oct}$ $\Rightarrow \uparrow$ URALS$_{Dec}$ linkage, which act on time scales that are inconsistent with the 2-month delay found in observations. For example, reduced sea ice may allow more heating of the atmosphere by the ocean to produce a Rossby wave train with an anomalous high over the Urals region (Honda et al., 2009), but this would be expected to manifest within a matter of days to a week. Alternatively, reduced ice may reduce local baroclinicity, which discourages cyclones from tracking into the Barents-Kara Seas and produces an anomalous high due to the relative absence of low-pressure systems (Inoue et al., 2012). This mechanism could introduce some delay between the ice perturbation and sea level pressure perturbation, but two months persistence of such a pattern is unlikely. Finally, reduced ice may increase atmospheric moisture content, leading to increased Eurasian snow cover, diabatic cooling and anomalously high sea level pressure over the continent (Liu et al., 2012; Cohen et al., 2014a; Garcia-Serrano and Frankignoul, 2014). Though this would plausibly lead to persistence on the required time scale, recent observational and modelling studies do not support a role for Eurasian snow in this teleconnection pathway (Kretschmer et al., 2016; Peings et al., 2017; Henderson et al., 2018), and we chose not to include it in our main analyses. Note that these mechanisms may still be responsible for contemporaneous forcing of the winter atmospheric circulation by winter sea ice variability, which has been suggested to be a stronger influence than the forcing by autumn sea ice variability (Blackport and Screen, 2019).

Lastly, our experience with the CEN offers some cautionary notes about its application to climate problems. The CEN approach was designed for hypothesis testing - that is, to test causal pathways that are thought or known to exist, either from

theory or existing evidence. It should not be used as an exploratory data analysis tool to search for causal pathways because the statistics behind the CEN do not know whether relationships are physically meaningful. One specific problem we encountered is that the algorithm may drop an existing causal linkage if a new variable is added. For example, when we introduce downward longwave radiation into the monthly CEN, its strong correlation with sea ice overrides the $\downarrow$ ICE$_{\text{Jan}}$ $\Rightarrow$$\uparrow$ URALS$_{\text{Feb}}$ linkage (see Fig. S9 compared to Fig. S1a). Since many climate variables are highly correlated, but not necessarily directly related via specific processes, the CEN's ability to identify physically meaningful linkages depends critically on the careful selection of input variables.

## 5   Concluding remarks

This study uses the Causal Effect Networks (CEN) framework to quantify the robustness of the stratospheric pathway between late autumn Barents-Kara sea ice and the February NAO, documenting its high level of intermittency in the observational record. The pathway has been relatively active over the satellite period, explaining approximately 26% of the interannual variability in the February NAO. However, this result is highly sensitive to which winters are included in the analysis. Results from a bootstrapping test show that the full stratospheric pathway appears in only 16% of the sample populations derived from the observations. The result reflects the strong internal variability of the midlatitude atmosphere and the likelihood that Arctic-midlatitude teleconnections may require certain background flow conditions. On synoptic time scales, we identify two-way interactions between Barents-Kara sea ice and the midlatitude circulation suggesting a role for atmospheric blocking over the Urals region and moist intrusions, both of which can reduce Barents-Kara sea ice. These synoptic processes can reinforce or interrupt the stratospheric pathway, contributing to intermittency. Finally, we cannot rule out that the causal linkages found on longer time scales may be artefacts of averaging over the synoptic processes, or even the result of entirely different mechanisms (Smith et al., 2017; Hell et al., 2019).

Coupled interactions between sea ice and the midlatitude circulation involve complicated lead-lag feedbacks over a range of time scales. Applying causal inference frameworks such as the CEN can help clarify some of the important physical processes at play, but in the end, models are required to improve our understanding. A complication is that the fidelity of climate models in representing the relevant processes is difficult to ascertain (King et al., 2016; Smith et al., 2017; Mori et al., 2019), especially those processes at fine spatial and temporal scales and their interactions across scales. But ways forward are indicated by this study, along with others (McCusker et al., 2016; Sun et al., 2016; Peings, 2019), that provide insight into which linkages are most robust, and which are subject to sampling issues within the relatively short observational record.

*Code availability.*   The codes to construct the CEN and figures in this study are available online at: https://github.com/petersiew/CEN

*Data availability.* ERA-Interim data are provided by European Centre for Medium-Range Weather Forecasts (ECMWF) online at :https: //www.ecmwf.int/en/forecasts/datasets/reanalysis-datasets/era-interim

*Author contributions.* PS conducted the analysis, prepared the figures, and wrote the paper with contributions from all co-authors. CL and
SPS conceived of the original idea. CL, SPS and MPK provided guidance on the interpretation of results.

*Competing interests.* Camille Li is a member of the editorial board of the journal.

*Acknowledgements.* This work was supported by the Research Council of Norway projects 255027 (DynAMiTe) and 272721 (Nansen
Legacy). We acknowledge the European Centre for Medium-Range Weather Forecasts for providing the ERA-Interim data. Finally, we thank
two anonymous reviewers for insightful suggestions and the Bjerknes storm tracks group for stimulating discussions, all of which that helped
improve this study.

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

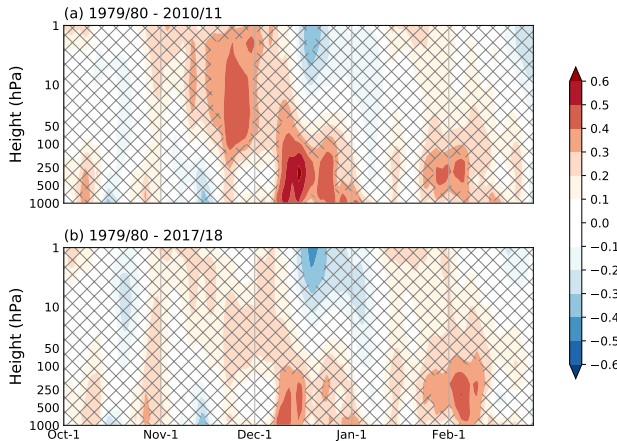

**Figure 1.** Lead-lag correlations (shading) between November Barents-Kara sea ice index (sign reversed) and polar cap height (70°N poleward) over the October-to-February cold season using ERA-Interim reanalysis for two periods: (a) 1979/80-2010/11 and (b) 1979/80-2017/18. Hatching indicates non-significant values at the 5% level using a two-tailed t-test. Linear trends and the seasonal cycle have been removed.

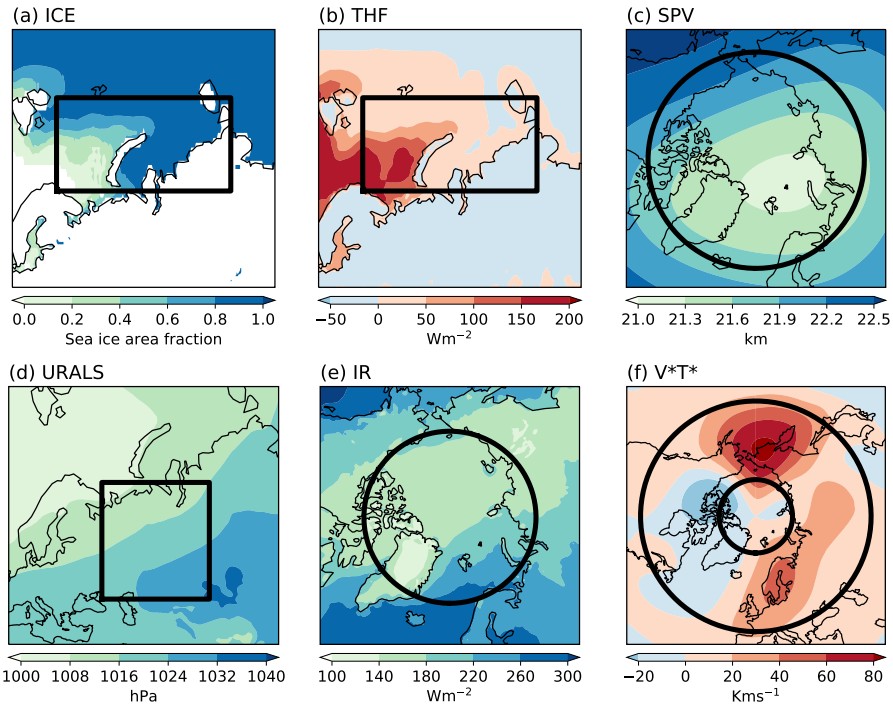

**Figure 2.** ERA-Interim (1979-2018) DJF climatologies (shading) of key variables and regions (black boxes) for computing area-averaged indices: (a) Sea ice area fraction, $70°$-$80°$ N, $30°$-$105°$ E, (b) Turbulent heat flux, $70°$-$80°$ N, $30°$-$105°$ E, (c) Stratospheric polar vortex, which is defined by 10-100 hPa geopotential height, $65°$-$90°$ N, (d) Urals sea level pressure, $45°$-$70°$ N, $40°$-$85°$ E, (e) Downward longwave radiation, $70°$-$90°$ N, (f) 100 hPa poleward eddy heat flux, $45°$-$75°$ N. For (b), turbulent heat flux from the ocean to the atmosphere is defined as positive. See section 2.1 for details.

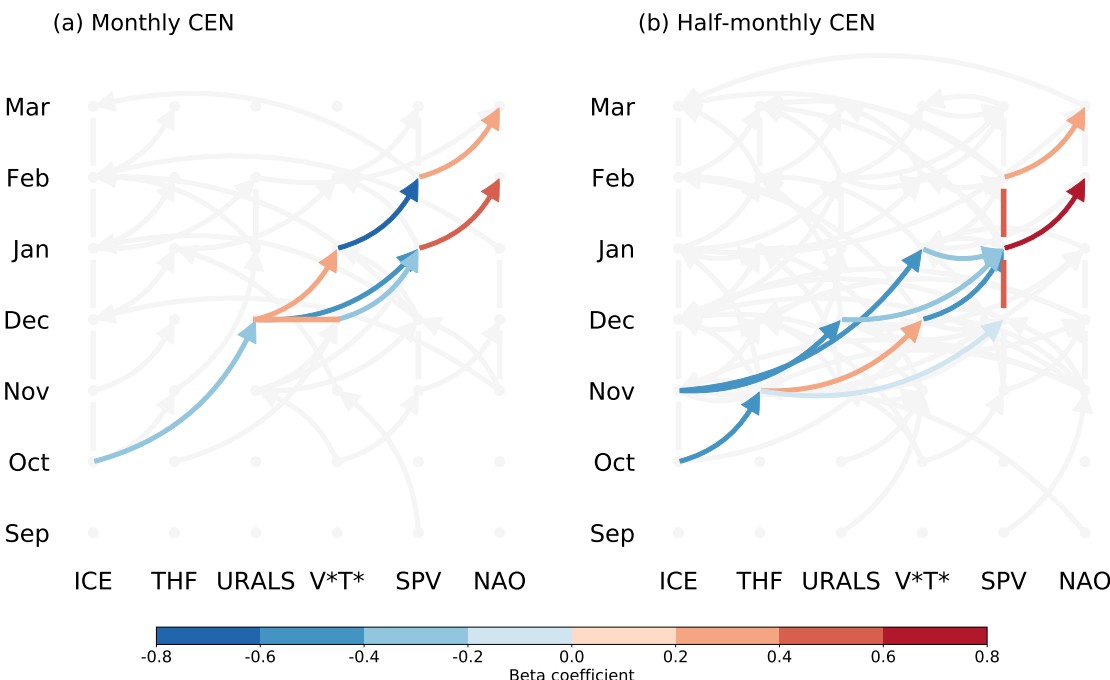

**Figure 3.** Seasonal evolution of the stratospheric pathway (indicated by coloured arrows) detected by the (a) monthly and (b) half-monthly CENs. Arrows indicate causal linkages; vertical lines indicate auto-correlation; horizontal bars indicate simultaneous relationships; colours show the sign and strength of the linkages as given by the CEN beta coefficients (see section 2.2). The grey background shows other significant linkages (arrows) and autocorrelations (vertical lines), but does not include simultaneous relationships. The half-monthly CEN in (b) has been aggregated into full months for ease of comparison with (a). See Fig. S2 for unaggregated version.

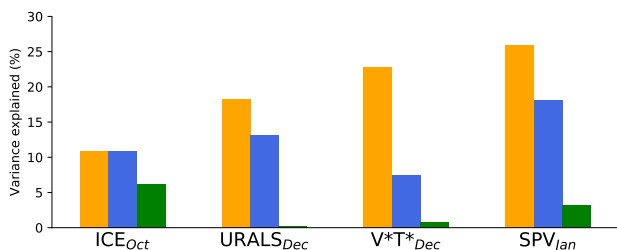

**Figure 4.** Explanatory power of the stratospheric pathway for the February NAO assessed via multiple linear regression. Orange bars show the cumulative variance explained when including each regressor in succession from left to right; blue bars show variance explained by the individual regressor; green bars show the reduction in total variance explained when removing that regressor. See section 3.2 for details.

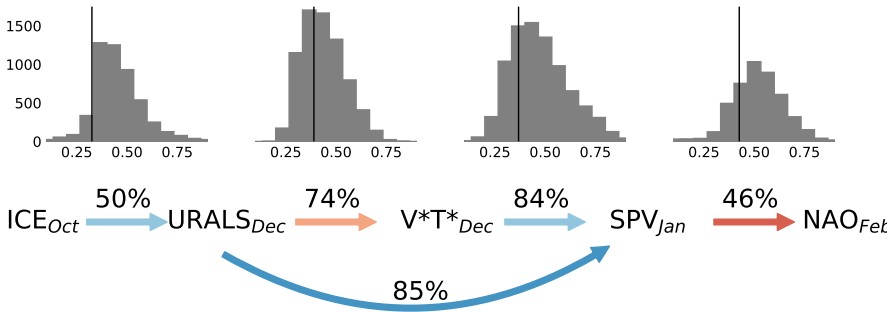

**Figure 5.** Results of a bootstrapping test to assess the robustness of causal linkages within the stratospheric pathway. Percentages above arrows show the occurrence rate of each linkage out of 10,000 bootstrap samples. Colours of the arrows (identical to Fig. 3) and the black lines show observed beta coefficients for each linkage for the reanalysis period. Histograms above show the corresponding distribution of beta coefficients (absolute value) in the bootstrap samples. The histogram for the ↑URALS$_{Dec}$ ⇒↓SPV$_{Jan}$ linkage is not shown. Note that the distributions are composed only of samples in which the linkage is detected.

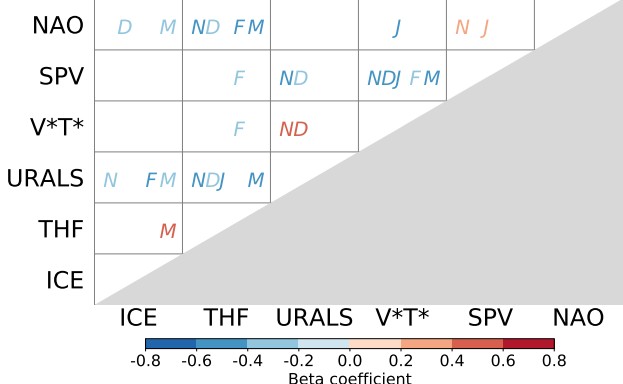

**Figure 6.** Simultaneous relationships between monthly indices in November (N), December (D), January (J), February (F) and March (M). Colours indicate the sign and strength of the relationship as given by the CEN beta coefficients (see section 2.2).

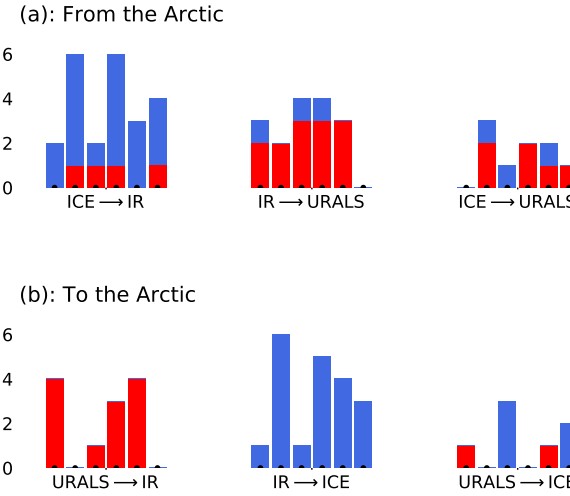

**Figure 7.** Results of the pentad CEN analysis assessing relationships between downward longwave radiation (IR), Barents-Kara sea ice (ICE) and Urals sea level pressure (URALS) aggregated into months (October, November, December, January, February and March from left to right). The height of each bar is the number of counts. (a) Linkages from the Arctic to the midlatitudes. (b) Linkages from the midlatitudes to the Arctic. Red (blue) colours denote positive (negative) relationships. See Fig. S8 for unaggregated version.

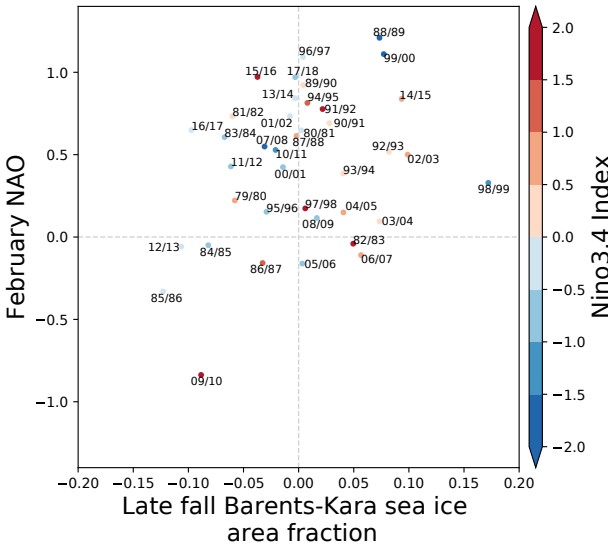

**Figure 8.** Scatter plots between February NAO and late fall (mean of October and November) Barents-Kara sea ice index for the reanalysis period. Shading indicates the DJF Nino3.4 index. Red (blue) denotes El Niño (La Niña) events.

**Table 1.** A summary of the casual effect of all ICE-NAO pathways. The $\downarrow ICE_{Oct} \Rightarrow \downarrow NAO_{Mar}$ pathway consists of both tropospheric and stratospheric branches.

| Pathway | Tropospehric | Stratospheric | Total |
|---|---|---|---|
| $\downarrow ICE_{Oct} \Rightarrow \downarrow NAO_{Feb}$ | N/A | 0.0823 | 0.0823 |
| $\downarrow ICE_{Oct} \Rightarrow \downarrow NAO_{Mar}$ | 0.0614 (70%) | 0.0258 (30%) | 0.0872 |
| $\downarrow ICE_{Jan} \Rightarrow \downarrow NAO_{Mar}$ | 0.137 | N/A | 0.137 |