# Peer review of "Intermittency of Arctic-midlatitude teleconnections: stratospheric pathway between autumn sea ice and the winter NAO"

_Weather and Climate Dynamics, 2019_

## Referee Comment (RC1) · Anonymous Referee #1 · 10 Dec 2019

The authors have examined in great detail the causal linkage between autumn Barents-Kara Sea ice anomalies and winter surface climate. The paper is well written and the logic is sound.

A piece of potentially useful information missing is why November is selected as the source timing and February as the resulting timing, given that October Barents-Kara Sea ice is even better correlated with January NAO variability than the November sea ice - February NAO counterparts. In addition, NAO variability is larger in January than in February, which makes better sense to explore the October-January pathway.

In addition, the sea ice-NAO linkage is sensitive to the investigation period, i.e., 1979-

2010 vs 1979-2018 as in Fig.1. There seem a few possibilities that might explain such non-robustness. Fore example, the time it takes from the sea ice to the NAO might vary in different periods (not always being three months). Alternatively, the timing might shift over time, e.g., due to changes in the seasonal cycle of the stratospheric jet strength in response to anthropogenic forcing.

---

## Referee Comment (RC2) · Anonymous Referee #2 · 12 Dec 2019

General Comments:

The present manuscript discusses the potential link between Barents and Kara sea ice in autumn and the late winter NAO via the so-called "stratospheric pathway. By applying a causal effect network (CEN) approach, the authors focus on the intermittency of the pathway and discuss the role of both sub-seasonal and synoptic processes. The manuscript is well written and structured and the results are presented in a balanced way and discussed in the context of the relevant literature. Overall, I find the analysis a valuable contribution to the ongoing debate about Arctic-mid-latitude linkages, highlighting the importance of different time-scales and potential non-stationarities as

well as the need for statistical concepts to deal with these challenges. I do have some comments which I think would improve the manuscript. In particular, the significance testing of the CEN analysis is not optimal and the regression analysis seems to lack some consistency with the CEN analysis. The results of the "Synoptic linkages and interactions across times scales" section on the other hand deserve more weight in my opinion as they are a novel attempt to explain non-stationarities of the stratospheric pathway and might be an important step towards reconciling model and observation analyses.

Specific Comments:

1) CEN method

I am well familiar with the applied method and am certainly convinced by its advantages and thus also pleased to see it being used. Nevertheless, I think that the authors do not apply it an optimal way. Since Kretschmer et al. (2016) and Runge et al. (2014) there have been several advancements to this approach as discussed in more detail in Runge et al. (2019, which was published after the author's submission) and Kretschmer et al (2018, npj) and Runge et al. (2018, Chaos). In particular the authors do not adequately account for the issue of multiple testing and false positives:

They only test links found in their step 1 (the correlation analysis) and then use partial correlation tests to check if the correlation from step 1 can be explained by confounders. Due to the involved multiple testing (in step 2) of links, the significance level alpha cannot be interpreted as the false positive rate. The authors, however, use this alpha level for further interpretations of their results. Note that this can be overcome by considering their step 1+2 only as a "condition selection step" and then testing each possible link again using partial correlations with the conditions identified before (as described in detail in Runge et al. 2019). Not only does this yield in more statistical interpretability but, furthermore, this can also lead to higher detection power. The reason is that a true link A->B can be overlooked because correlations are zero (see

their step 1) but conditioning out the influence of a common driver C (with opposite effects on A and B) reveals the actual signal (which might also affect the intermittency statistics). In this context, also note that estimating this condition set for each variable can be done by using different and rather liberal alpha levels (considered as a hyperparameter or a significance "threshold") and the "optimal" set can then be chosen, for example, based on the Akaike information criterion (AIC), as described in Runge et al. 2019.

Further, the authors should be aware that the overall resulting network is also subject to the "field significance" issue (see e.g. Wilks 2017, BAMS). This means that just by chance, some of the detected significant links will be false positives. This can also be addressed by applying false discovery rate corrections to the overall network. This issue might not be too relevant for the monthly CEN which only consists of few links, but might be an issue for the synoptic-scales CENs.

That said, I don't expect the authors to include all these novelties in their approach. Nevertheless, these issues should be discussed properly and the parts where referred to significance should be adopted. Further, I don't agree with their statement in l 136 that testing each link again in step 2 of the analysis in Kretschmer et al (2016) is "somewhat arbitrary". On the contrast, it is one way to deal with multiple testing problem as described above. Maybe the authors also want to consider comparing their results with the pcmci implementation provided here https://github.com/jakobrunge/tigramite/.

2) Strength of the pathway

When estimating the influence on NAO variability, the regression analysis seems somehow contradictory to the previous analysis. Why would one include the whole pathway if one believes that it represents an indirect chain of links? More precisely, if, for instance, the influence of BK SIC on SPV is via Urals then, in theory, the whole information is already contained in Urals and adding BK SIC in the regression should not provide additional information. Following the logic of a network

approach, the causal effect from A–>B is the sum over the products of link coefficients along all possible paths between the two variables (see for example here: https://github.com/jakobrunge/tigramite/blob/master/tutorials/tigramite_tutorial_causal_effects_mediation.ipynb )

Also, it seems obvious that adding more regressors to the regression analysis also increases its r2? Is the analysis performed for all years? Wouldn't it make sense to use a similar bootstrap approach or at least some leave-k-out cross validation?

As stated before, a more interesting analysis could be an attempt to quantify the contribution of sea ice to the NAO via the tropospheric and via the stratospheric pathways alone. This recent paper might me an interesting source of inspiration how do approach this using the CEN approach:

Saggioro, E. and Shepherd, T. G. (2019) Quantifying the timescale and strength of Southern Hemisphere intra-seasonal stratosphere-troposphere coupling. Geophysical Research Letters. ISSN 0094-8276

3) Intermittency

Addressing the intermittency of the Arctic – NAO link is very interesting and important given that it might provide an explanation why models and observation studies don't seem to agree on this topic. It is well known that not all extremely weak polar vortex states (or SSWs) affect tropospheric circulation (e.g. Karpechko et al. 2017, Runde et al. 2016). Do I understand the authors correctly that sea ice variability might also contribute to this downward intermittency (for example due to sub-seasonal-synoptic interactions)? I think it would make sense to highlight the intermittency of both the upward and the downward coupling mechanisms separately (with the upward part representing somehow the potential of sea ice to influence the NAO).

In my opinion the section "Synoptic linkages and interactions across times scales" is the most novel contribution to the rather large body of literature on this topic and deserves

to be highlighted even more. In this context it would be great if the difference between autumn/early and late winter linkages could be discussed in more detail. Which season of sea ice loss is most relevant for the stratospheric pathway, which for the tropospheric pathway. Is it possible to quantify how much of the intermittency is explained by things such as ENSO or synoptic variability?

Please also consider discussing/citing this recent paper: E Tyrlis, E Manzini, J Bader, J Ukita, N Hisahi, D Matei, Ural blocking driving extreme Arctic sea-ice loss, cold eurasia and stratospheric vortex weakening in autumn and early winter 2016-2017, Journal of Geophysical Research: Atmospheres 124, 11313-11329

Technical comments:

L3-4: not so obvious to me if really leading to a transition to –NAO (e.g. Karpechko et al 2017) L4: The Causal Effect Network... L32: Kretschmer et al. 2016 does not use lagged correlations L46: do these studies also consider BK sea ice? There is a lot of evidence that sea ice in the Pacific sector leads to a strengthening. L63: Maybe state which linkage you mean exactly. L189: 10,000 L 203: Why only those where it appears and not all? Fig. 3: switching the axes would make it more intuitive. L273-277: very interesting thoughts but should rather be moved to discussion?

References:

Runde, T., M. Dameris, H. Garny, and D. E. Kinnison, 2016: Classification of stratospheric extreme events according to their downward propagation to the troposphere. Geophys. Res. Lett., 43, 6665-6672, https://doi.org/10.1002/2016GL069569.

J. Runge, P. Nowack, M. Kretschmer, S. Flaxman, D. Sejdinovic, Detecting and quantifying causal associations in large nonlinear time series datasets. Sci. Adv. 5, eaau4996 (2019) https://advances.sciencemag.org/content/5/11/eaau4996

Kretschmer, M., J. Cohen, V. Matthias, J. Runge, D. Coumou (2018),The different stratospheric influence on cold-extremes in Eurasia and North America, npj Climate

and Atmospheric Science, doi:10.1038/s41612-018-0054-4

Causal network reconstruction from time series: From theoretical assumptions to practical estimation J Runge Chaos: An Interdisciplinary Journal of Nonlinear Science 28 (7), 075310

Karpechko, A. Y., Hitchcock, P., Peters, D. H. W., & Schneidereit, A. (2017). Predictability of downward propagation of major sudden stratospheric warmings. Quarterly Journal of the Royal Meteorological Society, 143( 704), 1459– 1470. https://doi.org/10.1002/qj.3017

Saggioro, E. and Shepherd, T. G. (2019) Quantifying the timescale and strength of Southern Hemisphere intra-seasonal stratosphere-troposphere coupling. Geophysical Research Letters. ISSN 0094-8276

D. S. Wilks, "The Stippling Shows Statistically Significant Grid Points": How Research Results are Routinely Overstated and Overinterpreted, and What to Do about It., https://doi.org/10.1175/BAMS-D-15-00267.1

E Tyrlis, E Manzini, J Bader, J Ukita, N Hisahi, D Matei, Ural blocking driving extreme Arctic sea-ice loss, cold eurasia and stratospheric vortex weakening in autumn and early winter 2016-2017, Journal of Geophysical Research: Atmospheres 124, 11313-11329

---

## Author Comment (AC1) · 7 Feb 2020

**Response to referees - Intermittency of Arctic-midlatitude teleconnections: stratospheric pathway between autumn sea ice and the winter NAO**

Peter Yu Feng Siew, Camille Li, Stefan Pieter Sobolowski, and Martin Peter King

**Anonymous Referee #1**

*The authors have examined in great detail the causal linkage between autumn Barents-Kara Sea ice anomalies and winter surface climate. The paper is well written and the logic is sound.*

We would like to thank the reviewer for the comments. We have provided a point-by-point response to the comments below. Reviewers' comments are in blue, our replies are in black.

1. *A piece of potentially useful information missing is why November is selected as the source timing and February as the resulting timing, given that October Barents-Kara Sea ice is even better correlated with January NAO variability than the November sea ice - February NAO counterparts. In addition, NAO variability is larger in January than in February, which makes better sense to explore the October-January pathway.*

   In the CEN analysis, we do not select specific pathways between specific months, but rather allow the CEN algorithm to objectively identify causal linkages that make up pathways. In the monthly CEN (Fig. 2a and Fig. S1a in the submitted manuscript), the stratospheric pathway does in fact originate with October Barents-Kara sea ice. In the half-monthly CEN (Fig. 2b and Fig. S1b in the submitted manuscript), a pathway from November Barents-Kara sea ice to January NAO does appear (consistent with some previous studies), although it does not go through the stratosphere, which is the focus of this study. Fig. 1 shows simple lagged correlations between October and November Barents-Kara sea ice indices, and NAO from October to February. We can see that the $ICE_{Oct}$-$NAO_{Feb}$ and $ICE_{Nov}$-$NAO_{Jan}$ correlations are the strongest, and it is in fact these pathways that the CEN picks. There are also quite strong correlations between ICE and December NAO, but these are not part of any connected pathways identified by the CEN analysis. This could be due to the fact that these correlations are not stationary in time, and depend on the exact reanalysis period used (also discussed in Introduction of submitted manuscript and mentioned below in point #2).

[Figure]

Fig. 1: The correlation between October (red) and November (blue) Barents-Kara sea ice indices, and NAO from October to February. Outer grey lines show significant correlations at a 5% level using a two-tailed t-test.

2. In addition, the sea ice-NAO linkage is sensitive to the investigation period, i.e., 1979-2010 vs 1979-2018 as in Fig.1. There seems to be a few possibilities that might explain such non-robustness. For example, the time it takes from the sea ice to the NAO might vary in different periods (not always being three months). Alternatively, the timing might shift over time, e.g., due to changes in the seasonal cycle of the stratospheric jet strength in response to anthropogenic forcing.

We agree that there are a number of factors that may contribute to the non-robustness of the linkage. The non-stationarity of the pathway's timing certainly might be one of them. While a comprehensive investigation into all the possible mechanisms is beyond the scope of the paper, we do make an effort to discuss what we feel are some prime candidates.

In the revised manuscript, we will expand the discussion, including a mention of the timing issue raised here, and also mention the influence of anthropogenic forcing on the seasonal cycle of the stratosphere as possible sources of intermittency.

**Anonymous Referee #2**

General Comments: The present manuscript discusses the potential link between Barents and Kara sea ice in autumn and the late winter NAO via the so-called "stratospheric pathway. By applying a causal effect network (CEN) approach, the authors focus on the intermittency of the pathway and discuss the role of both sub-seasonal and synoptic processes. The manuscript is well written and structured and the results are presented in a balanced way and discussed in the context of the relevant literature. Overall, I find the analysis a valuable contribution to the ongoing debate about Arctic-mid-latitude linkages, highlighting the importance of different time-scales and potential non-stationarities as well as the need for statistical concepts to deal with these challenges. I do have some comments which I think would improve the manuscript. In particular, the significance testing of the CEN analysis is not optimal and the regression analysis seems to lack some consistency with the CEN analysis. The results of the "Synoptic linkages and interactions across times scales" section on the other hand deserve more weight in my opinion as they are a novel attempt to explain non-stationarities of the stratospheric pathway and might be an important step towards reconciling model and observation analyses.

We would like to thank the reviewer for the supportive comments and suggestions for improvement. We have provided a point-by-point response to the specific comments below. Reviewers' comments are in blue, our replies are in black.

1. I am well familiar with the applied method and am certainly convinced by its advantages and thus also pleased to see it being used. Nevertheless, I think that the authors do not apply it an optimal way. Since Kretschmer et al. (2016) and Runge et al. (2014) there have been several advancements to this approach as discussed in more detail in Runge et al. (2019, which was published after the authors submission) and Kretschmer et al (2018, npj) and Runge et al. (2018, Chaos). In particular the authors do not adequately account for the issue of multiple testing and false positives: They only test links found in their step 1 (the correlation analysis) and then use partial correlation tests to check if the correlation from step 1 can be explained by confounders. Due to the involved multiple testing (in step 2) of links, the significance level alpha cannot be interpreted as the false positive rate. The authors, however, use this alpha level for further interpretations of their results. Note that this can be overcome by considering their step 1+2 only as a "condition selection step" and then testing each possible link again

using partial correlations with the conditions identified before (as described in detail in Runge et al. 2019). Not only does this yield in more statistical interpretability but, furthermore, this can also lead to higher detection power. The reason is that a true link A->B can be overlooked because correlations are zero (see C2 WCDD Interactive comment Printer-friendly version Discussion paper their step 1) but conditioning out the influence of a common driver C (with opposite effects on A and B) reveals the actual signal (which might also affect the intermittency statistics). In this context, also note that estimating this condition set for each variable can be done by using different and rather liberal alpha levels (considered as a hyperparameter or a significance "threshold") and the "optimal" set can then be chosen, for example, based on the Akaike information criterion (AIC), as described in Runge et al. 2019. Further, the authors should be aware that the overall resulting network is also subject to the "field significance" issue (see e.g. Wilks 2017, BAMS). This means that just by chance, some of the detected significant links will be false positives. This can also be addressed by applying false discovery rate corrections to the overall network. This issue might not be too relevant for the monthly CEN which only consists of few links, but might be an issue for the synoptic-scales CENs. That said, I don0 t expect the authors to include all these novelties in their approach. Nevertheless, these issues should be discussed properly and the parts where referred to significance should be adapted. Further, I don't agree with their statement in l 136 that testing each link again in step 2 of the analysis in Kretschmer et al (2016) is "somewhat arbitrary". On the contrast, it is one way to deal with multiple testing problem as described above. Maybe the authors also want to consider comparing their results with the pcmci implementation provided here https://github.com/jakobrunge/tigramite/

Thank you for informing us about the latest publications and advances in the CEN approach. We agree that the CEN algorithm used in the submitted manuscript is not complete. We tested many of the suggested CEN modifications and redid our analyses accordingly - while the main results do not change, there are a number of aspects detailed here that we feel are worth including in a revised manuscript.

We have now adopted two modifications to improve the CEN method. Firstly, the Akaike information criterion (AIC) is used to select the optimal condition set of each variable in the PC step (following terminology in Runge et al. 2019), with significance levels set at 0.05, 0.1 and 0.2. Secondly, the MCI test is introduced, whereby we test each possible link again using partial correlations with the conditions identified in the PC step.

Fig. 3 shows the stratospheric pathway in the (a) monthly and (b) half-monthly CEN using the modified algorithm. Some new linkages appear as a result of the higher detection power with the MCI step. The monthly CEN detects an additional branch of the stratospheric pathway which links $URALS_{Dec}$ to $NAO_{Mar}$. In the half-monthly CEN, a few more linkages (e.g., $THF_{Nov} \rightarrow SPV_{Dec}$ and $V^*T^*_{Jan} \rightarrow NAO_{Feb}$) are detected, but they do not change the results in terms of overall pathways. The intermittency rates of individual segments in the original $ICE_{Oct} \rightarrow NAO_{Feb}$ pathway do not change significantly (Fig. 3). The full stratospheric pathway now appears in 16.5% (compared to 15% before) of bootstrap samples.

In addition, we performed several other sensitivity tests on the algorithm:
- In the PC step, we used the Akaike Information criterion (AIC) to select the optimal significance level for individual variables in individual months, with three significance level sets: 0.05, 0.1, 0.2 (Fig. 2); 0.05, 0.1, 0.2, 0.3 (Fig. 4a); 0.05, 0.1, 0.2, 0.3, 0.4, 0.5 (Fig. 4b). The stratospheric pathways in the monthly CEN are unchanged in all sets. The pathways in the half-monthly CEN change slightly using the second and third sets. With the inclusion of less strict significance levels at which (partial) correlations are deemed

not significant, more parents are identified in the PC steps. This leads to higher dimensionality in the MCI step, which means that causal linkages are rejected more easily (e.g., disappearance of the $SPV_{Jan} \rightarrow NAO_{Feb}$ in Figs. 4a & 4b). Because our focus is the monthly CEN, we prefer to use the significance set 0.05, 0.1, 0.2, which produces consistent pathways in the half-monthly CEN as well, but we will discuss the sensitivity of the half-monthly results to this analysis choice.

- We performed a quick test applying the Hochberg-Benjamini false discovery rate (FDR) control to adjust the p-values in the MCI test, although the reviewer felt that this refinement was less important than others. This approach greatly reduces the number of detected linkages (Fig. 5). One problem we had was in determining the number of independent tests m to calculate the adjusted P-value = P*m/r, where P is the original P-value and r is the rank of the original P-value (sorted in ascending order). We used m=360 here (5 months * 6 variables* 6 variables * 2 lags), but these 360 tests are not truly independent, given the strong autocorrelation in many of these variables. We are not able to find a straightforward way to deal with this, and are inclined to leave out this FDR step.

In the revised manuscript, we will modify the Methods section to describe the modifications to the algorithm, primarily the MCI and the AIC. Results and the discussion will be revised to account for the new results. Since the conclusions drawn from the new results are consistent with the previous results, there will not be any significant changes to the main messages. We do agree with the reviewer that the phrase "somewhat arbitrary" in L136 was poorly worded. We will delete this sentence since we now include the MCI test.

[Figure]

Fig.2 : As in Fig. 3 in the submitted manuscript, but using the modified CEN algorithm (with the addition of the AIC with significance set 0.05, 0.1, 0.2 and the MCI test).

[Figure]

Fig. 3: As in Fig. 4 in the submitted manuscript, but using the modified CEN algorithm.

[Figure]

Fig. 4a: As in Fig. 2, but using AIC with significance set 0.05, 0.1, 0.2, 0.3.

[Figure]

Fig. 4b: As in Fig. 2, but using AIC with significance set 0.05, 0.1, 0.2, 0.3, 0.4, 0.5.

[Figure]

Fig. 5: As in Fig. 2a, but with the application of the False Discovery Rate correction using m=360.

2. Strength of the pathway. When estimating the influence on NAO variability, the regression analysis seems somehow contradictory to the previous analysis. Why would one include the whole pathway if one believes that it represents an indirect chain of links? More precisely, if, for instance, the influence of BK SIC on SPV is via Urals then, in theory, the whole information is already contained in Urals and adding BK SIC in the regression should not provide additional information. Following the logic of a network, the causal effect from A–>B is the sum over the products of link coefficients along all possible paths between the two variables (see for example here: https://github.com/jakobrunge/tigramite/blob/master/tutorials/tigramite_tutorial_causal_effects_mediation.ipynb ). Also, it seems obvious that adding more regressors to the regression analysis also increases its r2? Is the analysis performed for all years? Wouldn't it make sense to use a similar bootstrap approach or at least some leave-k-out cross validation? As stated before, a more interesting analysis could be an attempt to quantify the contribution of sea ice to the NAO via the tropospheric and via the stratospheric pathways alone. This recent paper might me an interesting source of inspiration how do approach this using the CEN approach: Saggioro, E. and Shepherd, T. G. (2019) Quantifying the timescale and strength of Southern Hemisphere intra-seasonal stratosphere-troposphere coupling. Geophysical Research Letters. ISSN 0094-8276

Thank you for these comments. One purpose of this analysis is to carry out a simple assessment of the strength of the pathway as observed within the whole satellite period, using all years (1979-2018) without resampling. This analysis shows that the pathway can explain 26% of the interannual variability in $NAO_{Feb.}$ We agree that it is a bit confusing to introduce this information at this point of the manuscript. We suggest to streamline the material, and move it earlier in the manuscript - right after the identification of the pathway and before the intermittency test to achieve better logical flow.

In response to the reviewer's other concerns/questions: Yes, adding more regressors will usually increase the variance explained. Our purpose here was to compare this to the other bars in

Fig. 5 in the submitted manuscript, to explore how different regression approaches (blue bars showing the effect of individual regressors versus green bars showing the effect of removing individual regressors) gives slightly different views on how the information is passed through the pathway.

As the reviewer suggests, we can quantify the relative contribution of the tropospheric and stratospheric ICE-NAO pathways as in Runge et al. 2015. For example, $ICE_{Oct}$-$NAO_{Mar}$ (Fig. 6) consists of both tropospheric and stratospheric components. The tropospheric pathway has a mediated causal effect: -0.661 ($ICE_{Oct} \rightarrow THF_{Nov}$) * 0.384 ($THF_{Nov} \rightarrow URALS_{Jan}$) * 0.517 ($URALS_{Jan} \rightarrow URALS_{Feb}$) * -0.468 ($URALS_{Feb} \rightarrow NAO_{Mar}$) = 0.0614; The stratospheric pathway has a mediated causal effect: -0.326 ($ICE_{Oct} \rightarrow URALS_{Dec}$) * 0.368 ($URALS_{Dec} \rightarrow V*T*_{Jan}$) * -0.715 ($V*T*_{Jan} \rightarrow SPV_{Feb}$) * 0.301 ($SPV_{Jan} \rightarrow NAO_{Mar}$) = 0.0258. Therefore, the tropospheric and stratospheric pathways account for 70.4% (0.0614/(0.0614+0.0258)) and 29.6% (0.0258/(0.0614+0.0258)) of the total causal effect, respectively. It is interesting that the tropospheric pathway is stronger, a point that, to our knowledge, has not been demonstrated previously. The result is perhaps unsurprising, given that the stratospheric pathway is undermined by both noise from the troposphere and noise from the stratosphere. Other pathways exist in the CEN, but we note that it is only meaningful to compare pathways with the same timing in terms of initial and ending months. Saggioro and Shepherd (2019) indeed provide a simple prediction model to quantify the strength of links, but it might be too complicated to apply this in our CEN since we are exploring pathways with multiple linkages.

In the revised manuscript, we will move this material into the end of section 3.1 (before the intermittency analysis), state clearly the purpose of these analyses, and streamline the text. We will also include an evaluation of the relative strength of pathways in terms of the mediated and total causal effects, as described above.

[Figure]

Fig. 6: As in Fig. 2a, but highlighting the $ICE_{Nov} \rightarrow NAO_{Mar}$ stratospheric and tropospheric pathways.

3. Intermittency. Addressing the intermittency of the Arctic – NAO link is very interesting and important given that it might provide an explanation why models and observation studies don't seem to agree on this topic. It is well known that not all extremely weak polar vortex states (or SSWs) affect tropospheric circulation (e.g. Karpechko et al. 2017, Runde et al. 2016). Do I understand the authors correctly that sea ice variability might also contribute to this downward intermittency (for example due to sub-seasonal-synoptic interactions)? I think it would make sense to highlight the intermittency of both the upward and the downward coupling mechanisms separately (with the upward part representing somehow the potential of sea ice to influence the NAO).

Yes, in this section, overall we are trying to make a point that there is quite some intermittency along the entire pathway that could contribute to the disagreement between models and observations, and that this intermittency arises from processes operating at different time scales.

Ice certainly plays a role in the intermittency of these pathways. Fig. 7 in the submitted manuscript breaks the synoptic Arctic-midlatitude interactions into "from Arctic" and "to Arctic" linkages, both of which focus on the upward coupling. The "from Arctic" case primarily shows the linkages $\downarrow ICE \rightarrow \uparrow IR \rightarrow \uparrow URALS$. In this case, synoptic processes reinforce the upward coupling seen in the monthly and half-monthly CEN. The "to Arctic" case exhibits evidence of large-scale flow conditions favouring moist intrusions or simply moisture transport by cyclones tracking into the Barents ($\uparrow URALS \rightarrow \uparrow IR \rightarrow \downarrow ICE$), which would act as a positive feedback on ice reductions. This means that even if there is low autumn sea ice in a given year, synoptic internal variability (which is large) could create anomalously low SLP over the Urals, leading to negative downward IR anomalies (lack of moist intrusions/cyclones) and positive ice anomalies, thus interrupting the upward coupling.

Pinning down the "source" of the intermittency is actually quite an interesting question. In the $ICE_{Oct} \rightarrow NAO_{Feb}$ stratospheric pathway in the monthly CEN, the upward coupling includes the segments $ICE_{Oct} \rightarrow URALS_{Dec}$ and $URALS_{Dec} \rightarrow V^*T^*_{Dec}$, whose intermittency rates are 50% and 74%, respectively (Fig. 3). The occurrence of these two linkages together is seen in about 41% out of 10,000 bootstrap samples, meaning that most of the time when we have the $ICE_{Oct} \rightarrow URALS_{Dec}$ linkage, the subsequent link to $V^*T^*_{Dec}$ is also seen. Conversely, only about half of the time that the linkage $URALS_{Dec} \rightarrow V^*T^*_{Dec}$ is detected is it preceded by the $ICE_{Oct} \rightarrow URALS_{Dec}$ linkage. It seems most likely that most the intermittency in both segments (individually and in terms of their "combined" occurrence rate) stems from variability in $URALS_{Dec}$ related to atmospheric internal variability.

In the revised manuscript, we will provide a more comprehensive discussion of how to interpret the intermittency analysis, and contrast the roles of different processes in the various pathways connecting ICE to NAO (Fig. 2). The role of ice in the downward intermittency is a bit trickier to assess and is likely a study in its own right. Certainly stratospheric internal variability will contribute to this but also sub-seasonal sea ice variability may enhance/mitigate downward signals. We will include some additional discussion on this topic based on previous studies such as those mentioned by the reviewer.

4. In my opinion the section "Synoptic linkages and interactions across times scales" is the most novel contribution to the rather large body of literature on this topic and deserves to be highlighted even more. In this context it would be great if the difference between autumn/early and late winter linkages could be discussed in more detail. Which season of sea ice loss is most relevant for the stratospheric pathway, which for the tropospheric pathway. Is it possible

to quantify how much of the intermittency is explained by things such as ENSO or synoptic variability? Please also consider discussing/citing this recent paper: E Tyrlis, E Manzini, J Bader, J Ukita, N Hisahi, D Matei, Ural blocking driving extreme Arctic sea-ice loss, cold eurasia and stratospheric vortex weakening in autumn and early winter 2016-2017, Journal of Geophysical Research: Atmospheres 124, 11313-11329

We agree that the synoptic CEN could be expanded to provide more insight into the roles of these processes throughout the cold season. Fig. 7a shows the full synoptic CEN with IR (downward longwave radiation), URALS and ICE, and Fig. 7b shows the CEN-aggregated over six individual months (O, N, D, J, F and M) through the cold season. In general these synoptic linkages show no clear trend evolving from fall to winter but strong variability between individual months. Synoptic linkages with a higher persistence might be seen on longer timescales. For example, an extensive amount of synoptic linkages ↓ICE→↑IR→↑URALS in November and January might help to form the $ICE_{Nov}$→$URALS_{Dec}$ in the half-monthly (Fig. 2b) and $ICE_{Jan}$→$URALS_{Feb}$ in monthly CEN (Fig. 2a). Similarly, a large amount of synoptic linkages ↑URALS→↑IR in October and ↑IR→↓ICE in November can reinforce the $ICE_{Oct}$→$URALS_{Dec}$ linkages in the monthly CEN. This might also explain why the ICE-to-URALS linkage is not found in December as there is only one synoptic linkage ↑URALS→↑IR→↓ICE found in December. The Tyrlis et al. (2019) study about Urals blocking leading to sea ice reduction and subsequent negative NAO conditions via sudden stratospheric warmings is very relevant, and will certainly be brought into the discussion.

As for the El Nino-Southern Oscillation (ENSO), previous work has shown that both ENSO (Figs 5e and 5f in King et al. 2018) and Barents-Kara sea ice (Fig. 5e in King et al. 2016) are associated with the "Scandanivan pattern" mode of variability characterised by high pressure anomalies over the Urals. This suggests that ENSO can potentially interfere with the ICE-NAO stratospheric - for example, in a cold season with low autumn sea ice, an El Niño (La Niña) could reinforce (weaken) the pathway, as discussed in the Discussion section of the submitted manuscript. However, quantitatively, it is tricky to assess the contribution of ENSO to the intermittency with such a short observational record. Moreover, the association between ENSO and the winter NAO/NAM is rather weak (Brönnimann 2007, Domeisen et al., 2019). This can also be seen in Fig. S7 of the submitted manuscript, which shows high (red) and low (blue) Nino3.4 values on both the lower (negative NAO) and upper (positive NAO) sides of the y-axis.

In the revised manuscript, we will expand the discussion of the synoptic linkages and time scale interactions in greater detail, and include a version of Fig. 7b showing the differences between different months. Moreover, we will move Fig. S7 into the main manuscript and expand the discussion as indicated above. In the study we hypothesize that the synoptic variability and ENSO are some possible sources of interference that might lead to intermittency, and show supporting evidence from the CEN. However, we do not provide an in-depth analysis of specific cases of interference, or how the interference occurs. While an interesting investigation in its own right, this is beyond the scope of this paper and is left for future work.

[Figure]

Fig. 7a: As in Fig. S6 in the submitted manuscript, but from the modified CEN algorithm and showing the causal links through the extended ONDJFM cold season (one more month added to the NDJFM cold season used previously).

[Figure]

Fig. 7b: As in Fig. 7 in the submitted manuscript, but aggregated by individual months (bars for each linkage are O, N, D, J, F, M from left to right)

5. Technical comments: L3-4: not so obvious to me if really leading to a transition to –NAO (e.g. Karpechko et al 2017)

It is a good point that the downward coupling following polar vortex weakening is not limited to the European and Atlantic sector. We will mention the AO signature as well.

L4: The Causal Effect Network. . .

Thank you, this will be changed.

L32: Kretschmer et al. 2016 does not use lagged correlations

The reference will be removed from that sentence.

L46: do these studies also consider BK sea ice? There is a lot of evidence that sea ice in the Pacific sector leads to a strengthening

Some of these experiments involved pan-Arctic sea ice reductions instead of Barents-Kara sea ice only. The statement was a general comment alluding to the fact that not all sea ice removal experiments agree with the proposed NAO/AO- response. To be more precise, we will clarify that reduction of sea ice in Pacific can lead to strengthening polar vortex and thus an opposite signed response (Sun et al. 2015).

L63: Maybe state which linkage you mean exactly.

We refer here to the linkage between sea ice loss and Eurasian cooling. We will state it more clearly.

L189: 10,000

Thank you, we will change it.

L 203: Why only those where it appears and not all?

Thank you. We think the reviewer was asking why the distributions are composed only of samples in which the linkage appears. The reason is that the beta coefficient can only be calculated when the CEN detects a linkage in a bootstrap sample, so only those samples are included.

Fig. 3: switching the axes would make it more intuitive.

Thank you for your suggestion. We will play around with switching the axes for the revised manuscript.

L273-277: very interesting thoughts but should rather be moved to discussion

Thank you. We will move this content to the Discussion section.

**References:**

Brönnimann, S. "Impact of El Niño–southern oscillation on European climate." *Reviews of Geophysics* 45.3, 2007

Domeisen, D. I., Garfinkel, C. I., and Butler, A. H.: The teleconnection of El Niño Southern Oscillation to the stratosphere, Reviews of Geophysics, 57, 5–47, 2019

King, M. P., Hell, M., and Keenlyside, N.: Investigation of the atmospheric mechanisms related to the autumn sea ice and winter circulation link in the Northern Hemisphere, Climate dynamics, 46, 1185–1195, 2016

King, M. P., Herceg-Bulić, I., Kucharski, F., and Keenlyside, N.:. Interannual tropical Pacific sea surface temperature anomalies teleconnection to Northern Hemisphere atmosphere in November. *Climate dynamics*, *50*(5-6), 1881-1899, 2018

Sun, L., Deser, C., and Tomas, R. A.: Mechanisms of stratospheric and tropospheric circulation response to projected Arctic sea ice loss,Journal of Climate, 28, 7824–7845, 2015

Runge, Jakob, Vladimir Petoukhov, Jonathan F. Donges, Jaroslav Hlinka, Nikola Jajcay, Martin Vejmelka, David Hartman, Norbert Marwan, Milan Paluš, and Jürgen Kurths: "Identifying Causal Gateways and Mediators in Complex Spatio-Temporal Systems." Nature Communications 6: 8502, 2015

Runge, P. Nowack, M. Kretschmer, S. Flaxman, D. Sejdinovic, Detecting and quan-tifying causal associations in large nonlinear time series datasets. Sci. Adv. 5,eaau4996 (2019)

Saggioro, Elena, and Theodore G. Shepherd. "Quantifying the Timescale and Strength of Southern Hemisphere Intraseasonal Stratosphere-troposphere Coupling." Geophysical Research Letters 46.22, 2019

---

## Author Response (AR2)

**Response to minor revision - Intermittency of Arctic-midlatitude teleconnections: stratospheric pathway between autumn sea ice and the winter NAO**

Peter Yu Feng Siew, Camille Li, Stefan Pieter Sobolowski, and Martin Peter King
10 April 2020

**Anonymous Referee #1**

*The authors have addressed all my previous comments. Here're some suggested technical corrections:*

We would like to thank the reviewer for the comments again. We have provided a point-by-point response to the comments below.

1. L57 varibility -> variability
   Please see line 57.

2. L61 oppsite -> opposite
   Please see line 61.

3. L83 downard -> downward
   Please see line 83.

4. L019 refered -> referred
   Please see line 109.

**Anonymous Referee #2**

*This is my second review of the manuscript "Intermittency of Arctic-midlatitude teleconnections: stratospheric pathway between autumn sea ice and the winter NAO". The authors did a great job in addressing all of my comments and I much appreciate the overall thorough analysis and well written paper. In my opinion the manuscript is ready for publication. As outlined below, I only have a few minor comments left.*

We would like to thank the reviewer for the comments again. We have provided a point-by-point response to the comments below.

1. *Determining the number of independent tests m to calculate the adjusted P-value is indeed not trivial. Again, I don´t think it is a major issue here, so I agree with the authors that it can be ignored. But in case the authors are interested, I recommend this paper here: "Estimating and Controlling the False Discovery Rate for the PC Algorithm Using Edge-Specific P-Values", https://arxiv.org/abs/1607.03975. In this paper, it is suggested to take the maximum p-value for each link over all conditional independence tests, which is yet a very conservative estimate!*

Thank you for pointing out this paper. Thinking about the adjusted P-value has been an interesting exercise and has certainly made us think more about the meaning of our results.

2. L40-41: Maybe worth noting that this is consistent with Kolstad and Screen 2019. https://agupubs.onlinelibrary.wiley.com/doi/full/10.1029/2019GL083059

Kolstad and Screen 2019 has been mentioned in line 39 in this context.

3. Section 2.2. I think the authors should at some point mention that the algorithm used to construct the CEN is called PCMCI (Runge et al. 2019).

Yes, it's true that it would be useful for readers who wish to refer to previous literature on the method. We have now mentioned the name in line 110.

4. L 107: The wording "true causal driver" is a bit misleading.

We have deleted the word "true". Please see line 107.

5. Section 3.2 Is the Link from Urals_Dec to vT_Dec calculated from the contemporaneous link strength? That is, can it really interpreted here as the casual effect on a given path?

$URALS_{Dec} \rightarrow V^*T^*_{Dec}$ in the monthly CEN is indeed obtained from the simultaneous relationship. First, we note that there is a $URALS_{Dec} \rightarrow V^*T^*_{Jan}$ linkage in the monthly CEN (Fig. 3a), supporting the fact that there is indeed a causal effect of URALS on $V^*T^*$. Furthermore, we think the contemporaneous linkage can be interpreted as causal because it is also detected in the half-monthly CEN from the first half of December to the second half of December (Fig. S2). Together, these points indicate that there is some variability in the timing of the pathway that is not properly accounted for when using straight monthly means (related to the reviewer's point #6 below). Still, the monthly mean viewpoint can give useful information on causal effects when combined with the analyses at shorter time scales. Please see lines 208-210.

6. As the time-resolution and the relevant lags remain as an issue, maybe it should be made a bit clearer that this section describes only an attempt to estimate the total causal effect of the stratospheric pathway.

Please see lines 252-254.

7. L264: why is the physical interpretation unclear? I think it makes even sense that VT is not "needed" as a mediator if the pathway only takes a month and not two.

We meant to point out that this is not consistent with theory, which indicates that the way surface perturbations alter the polar vortex is by altering vertical wave activity flux from the troposphere to the stratosphere. Of course, as discussed before, it could very well be that there are slight variations in the timing of the connection through $V^*T^*$, which would then not show up in the CEN.

8. L279: The same is probably true for extremely strong SPV states.

We have included polar vortex strengthening events. Please see line 282.

9. L306: I don't understand this sentence.

Within an individual month, there are six pentads (e.g., days 1-5, 6-10, 11-15, 16-20, 21-25 and 26-28/30/31). For each pentad, the CEN allows the detection of causal linkages with a lag of one pentad or two pentads. Therefore, the maximum number of detected causal linkages for a given pair of variables within a month is 6*2=12.

10. L340 and following: Again, I think it makes sense to differentiate the intermittencies of the upward and of the downward stratospheric coupling and quantify both.

We have in fact differentiated between the upward and downward coupling processes in the revised manuscript, but perhaps did not clearly indicate that we did so in section 3.3, where it seemed to fit better. The upward coupling ($\downarrow ICE_{Oct} \rightarrow \uparrow URALS_{Dec} \rightarrow \uparrow V^*T^*_{Dec}$) and downward coupling ($\downarrow SPV_{Jan} \rightarrow \downarrow NAO_{Feb}$) processes have occurence rates of 41% and 46%, respectively (lines 275-281).

11. L358: Why "however"?

We have deleted the word "however". Please see line 361.

12. L372: Or SIC contributes to ENSO intermittency? The NAO can probably be seen as a collider of ENSO and SIC effect.

We agree that the stratospheric pathway initiated by sea ice also contributes to the intermittency of the pathway initiated by ENSO, or vice versa, but here we would like to focus on what factor contributes to the intermittency of the pathway initiated by sea ice.

13. L407 and following: I fully agree with the statement that CEN "depends critically on the careful selection of input variables". However, the claim that links disappear "if a new variable is added because of changes in the results of the partial correlations tests" seems a bit unfair to me because this effect is fully understood in a statistical sense: If the time-series are too similar, then they cancel out each other when testing for conditional independence. Therefore, choice of relevant data is crucial for CEN, as it is for any data analysis

We fully agree and see how the phase "because of changes in the results of the partial correlations tests" could be misleading. We do explain in the following sentences that this is well understood, and related to time series that are very similar. The misleading phrase has now been deleted (line 411).

[revised manuscript text omitted]